# Securing Fog Computing with a Decentralised User Authentication Approach Based on Blockchain

**DOI:** 10.3390/s22103956

**Published:** 2022-05-23

**Authors:** Otuekong Umoren, Raman Singh, Zeeshan Pervez, Keshav Dahal

**Affiliations:** School of Computing, Engineering and Physical Sciences, University of the West of Scotland, Paisley PA1 2BE, UK; otuekong.umoren@uws.ac.uk (O.U.); raman.singh@uws.ac.uk (R.S.); keshav.dahal@uws.ac.uk (K.D.)

**Keywords:** authentication, IoT, cloud computing, fog computing, blockchain technology, smart contracts

## Abstract

The use of low-cost sensors in IoT over high-cost devices has been considered less expensive. However, these low-cost sensors have their own limitations such as the accuracy, quality, and reliability of the data collected. Fog computing offers solutions to those limitations; nevertheless, owning to its intrinsic distributed architecture, it faces challenges in the form of security of fog devices, secure authentication and privacy. Blockchain technology has been utilised to offer solutions for the authentication and security challenges in fog systems. This paper proposes an authentication system that utilises the characteristics and advantages of blockchain and smart contracts to authenticate users securely. The implemented system uses the email address, username, Ethereum address, password and data from a biometric reader to register and authenticate users. Experiments showed that the proposed method is secure and achieved performance improvement when compared to existing methods. The comparison of results with state-of-the-art showed that the proposed authentication system consumed up to 30% fewer resources in transaction and execution cost; however, there was an increase of up to 30% in miner fees.

## 1. Introduction

The quick development in technology, Internet of Things (IoT) and edge computing has given room for corporations, government agencies, health services and businesses to provide their services through the cloud. Running these cloud services requires financial and computational resources to secure and protect data from malicious entities and to process large amounts of data generated through these services on edge and IoT devices. These data are considered vulnerable to breaches and need to be secure from malicious entities; this hereby makes authentication essential in IoT and cloud services [1].

The increase in cyber-attacks on various services has demonstrated the importance of secure and robust authentication systems. Notable attacks such as the Facebook breach in 2019 exposed the datasets and information of over 500 million users publicly. This included users’ Facebook IDs, names, and contact numbers [2]. In 2021 [3], a scraping method exploited LinkedIn’s API exposed data related to 700 million LinkedIn users that were shared on the dark web. Data harvested in such types of attacks contain contact numbers, email addresses, gender information, geolocation, and other social media details, thus having direct professional, social, and financial implications for the users and service providers. Yahoo in 2013 suffered a breach that had the information of 3 billion of its users accessed by a group of hackers [4], and in 2014, Yahoo suffered an attack in which names, email addresses, hashed passwords, dates of birth, and contact numbers were stolen. To make things worse, this was not made public until the stolen database was put up for sale in 2016 [4]. Zoom in 2020 [5] suffered an attack that had the login credentials of its users stolen; those stolen passwords were gathered through credential surfing, and successfully compromised accounts were packaged into a new database.

Authentication is considered a key factor that guarantees the security of data stored on cloud servers. Security is guaranteed through controlled access to data, services and resources. However, these authentication systems based on a centralised database or a trusted third party are vulnerable to cyber-attacks and breaches [6]. In IoT, devices generate a large quantity of data in relatively less time; these data are transferred from multiple IoT devices to the cloud through the internet simultaneously, sometimes causing network congestion. Fog computing was introduced to overcome the limitations related to IoT and the cloud. The fog computing environment operates between the cloud data centres and IoT devices. These fog devices and computing instances are commonly referred to as fog nodes, which are deployed with distributed ownership [7]. Authentication systems would benefit from the unique characteristics of fog computing such as the distribution of fog nodes, the decentralised storage of blockchain technology and smart contract, hereby ensuring secure authentication and mitigating cyber-attacks.

The introduction of fog computing provides a decentralised environment with the involvement of several fog devices to engage from different locations. This offers a solution for the enhancement of cloud computing services. Hence, data generated by edge or IoT devices are readily processed and passed to the cloud through the fog devices, hereby achieving its initial objective of off-loading computational load to edge devices, thus reducing communication overhead in the form of bandwidth and network congestion [1].

In fog computing, computation is pushed to edge devices, the characteristics of fog and edge devices such as low latency make it capable of processing information without the involvement of the cloud server [8]. However, the security challenges and limitations in fog computing includes trust, security and privacy due to its untrusted decentralised environment [1]. Fog computing still relies on a centralised entity (cloud) for authentication. The use of blockchain technology can help reduce this burden while saving computation and network cost, whilst achieving traceability. The unique characteristics of blockchain technology such as transparency, secure encryption, reliability, mutual authentication between nodes, the use of a distributed encrypted ledger, and distributed database in which operations are executed would be beneficial to fog-based authentication [9].

Various authentication systems use a centralised database, but these systems have limitations such as dependability on centralised authority and are vulnerable to attacks/security threats such as brute force, dictionary attack, and Man-in-the-Middle attack [6]. The use of a decentralised database utilising blockchain ledger has been proposed by different researchers [1,10,11].

This paper aims to propose a decentralised authentication system utilising blockchain technology and fog computing. It uses data types such as email address, Ethereum address, username, password and data from the biometric reader to register and authenticate users. With the characteristics and features of blockchain technology such as peer-to-peer, cryptography, consensus and the use of smart contracts, it addresses authentication using a decentralized database and peer-to-peer communication between fog devices (nodes). The proposed system achieves authentication and communication without a central authority.

The main contributions of this paper can be summarized as follows:Propose a secure decentralised user authentication that utilises blockchain technology, smart contract and secure ledger.The system can handle authentication requests using the username, password, Ethereum address, user email and data from a biometric sensor.The system must guarantee the non-transferability of user data extracted from a biometric reader.The system is scalable and can scale to multiple IoT devices.The proposed system utilises different authentication methods and data types for user registration and authentication, which will be an improvement when compared to existing methods.

The rest of this paper is organised as follows: Section 2 introduces the authentication systems, fog computing and blockchain technology. Section 3 discusses the related works. Section 4 presents the proposed authentication system. Section 5 provides the implementation details of the proposed system, followed by details of the experimental setup in Section 6. Section 7 presents the performance metrics and results. Section 8 presents the performance evaluation. Section 9 discusses the overall performance and security analysis of the proposed system. Section 10 concludes the paper along with future directions.

## 2. Background

This section presents an overview of authentication systems, fog computing and blockchain technology.

### 2.1. Authentication Systems

Authentication has become increasingly important in the world, with individuals, corporations, and businesses making use of it to control access to data and information. Basic authentication methods such as knowledge-based factors (username and password) are commonly and widely used across platforms and services, while other authentication mechanisms such as biometric authentication (inherent factors) and token-based authentication (possession factors) are increasingly becoming popular and used across services. Knowledge-based authentication systems utilise usernames and passwords (alphanumeric and special characters); these are considered conventional and are widely used across platforms and services. This popularity comes as a result of fast processing and ease of memorability. The popularity of password-based authentication systems makes them vulnerable to various attacks such as brute force, dictionary, shoulder surfing, and they have limitations such as character count, password length and strength of the password [12]. Graphical passwords are introduced to mitigate some vulnerabilities in text-based passwords. This method is resistant to attacks such as dictionary attacks because of its large password space [13].

Biometric authentication has been known to be an improvement on password-based authentication, hereby mitigating some of those vulnerabilities. It is usually grouped into physiological (fingerprints, DNA, face) and behavioural characteristics (keystroke, voice, signature, etc.). Face recognition and fingerprints have been largely used on smartphones, laptops, and other smart devices to control user access and for the purpose of authentication on services, bank applications, and social media. Biometric authentication also has limitations and threats; this is largely due to the immutability and management of biometric data in a centralised database or module. The disclosure of these biometric data can cause various security threats such as the reuse of stolen biometric data [14].

With the vulnerabilities and threats in password-based and biometric authentication, two-factor authentication is considered reliable and increases security in authentication [15]. It utilises both password-based and biometric authentication—the inclusion of token-based authentication systems to serve as an extra level of security in multiple or two-factor authentication systems [16]. However, the user is required to have and carry a physical or hardware authentication device or authentication software on a smartphone. Although these methods are popular, they are not widely used as the traditional username and password.

### 2.2. Fog Computing

The fog generally is described as a decentralised distributed computing system where several fog devices are owned by different entities, and organisations can engage from different locations such as hospitals, schools, airports, smart hubs, etc. Researchers [17] described fog computing as a virtualised environment that is not completely situated outside the network space and tasked with the delivery, storage, and computing of resources in cloud computing centres. These are various fog nodes with less storage and computing capabilities. Fog computing is widely considered as an extension of the cloud situated close to devices that collects data for resource-constrained IoT. These devices are called fog nodes having storage, network connection, and computational power and they are placed on different geographical locations with network connectivity.

The main characteristics of fog computing are described in [17] as given below:Adaptability: These are made up of numerous network sensors and other fog devices that deliver storage resources and handle computing tasks.Real-time communications: Fog computing gives real-time communication between fog nodes and with corresponding data deployed in the cloud.Physical distribution: Fog computing provides decentralised services and applications hosted on different locations.Less latency: Fog computing’s closeness to the edge devices reduces information computing time with the edge devices and aids position responsiveness to host fog devices on several locations.Compatibility: Fog modules are designed to work with several platforms through many service providers.Web analytics and cloud integration: The fog’s position between the cloud and the edge devices is vital to the computing and processing of data close to the edge devices.Heterogeneity: Fog nodes and edge devices are made by different organisations and have different features, and it is necessary they are hosted according to their features.

### 2.3. Blockchain Technology

Blockchain is a distributed ledger technology that utilises consensus algorithms and cryptographic techniques to attain features such as immutability, traceability, anonymity, security, transparency and decentralisation. On the blockchain, confirmed transactions are validated by all nodes and recorded with a timestamp. Transparency is provided by the distribution of the ledger between the members of the blockchain network. Security of the data recorded in the ledger is guaranteed and cannot be tampered [18]. The smart contract is one of the important features in blockchain; it is a short-code included in the blockchain and can be automatically executed when the conditions are met. There are three known classes of blockchain: these are private, public and consortium blockchain. A private blockchain is controlled by a single organisation, governing the activities in the blockchain such as those who can participate in the blockchain and is considered more trusted among participants. A public blockchain is permissionless, and any entity can participate in it. Its highly decentralised nature brings about drawbacks such as privacy, security and performance. A consortium blockchain is a permissioned blockchain built by a consortium with several entities. Entities in this type of blockchain are nodes. The consortium controls which entity or organisations participate in the blockchain [18].

Blockchain technology has increasingly gained popularity in a variety of applications such as auctions [10], mutual authentication, and IoT [1] which has been the main beneficiary of blockchain technology’s features such as the decentralised characteristics which allow devices to connect and share sensitive data securely in an IoT environment. These attributes enable blockchain to provide the underlying secure storage and a trustworthy computational platform for authentication systems. Devices in a blockchain network establish trust through a consensus algorithm; this enables devices to keep a digital signature and cryptographic keys (private and public keys). This aids the communication and transaction between devices in the network; the immutability of these transactions makes them tamper-proof on the blockchain network.

## 3. Related Works

There have been several works proposed on authentication which have utilised fog computing and blockchain technology in IoT, auctions, cloud computing and other systems mainly in decentralised methods. In this section, a review has been done on schemes that included authentication, fog computing, blockchain technology, IoT and cloud computing to understand the proposed scheme.

In [19], the authors proposed a solution with the aim of solving the problems accompanied by a single root of trust in a distributed management of identity and authorisation policies using blockchain technology. This solution was integrated within FIWARE (an open-source technology used for the development of digital twins, data spaces and smart solutions) and was made as an authentication and authorisation solution for smart cities. The system was integrated with its set of security policies, which required other systems to be aware of these policies to permit the use of the same credentials as a single sign-on across other infrastructure.

A decentralised authentication and access control system for lightweight IoT devices was proposed by Khalid et al. [20]. This system utilised the advantages of fog computing, the distributed nature, peer-to-peer, and immutability of blockchain technology. This enables secure communications between IoT devices in different systems [20]. This system was limited to secure communications between IoT devices through blockchain technology’s peer-to-peer properties.

In [1], Patwary et al. proposed a decentralised mutual authentication using blockchain technology that authenticates devices independently without trusted third parties. This was designed to use an Ethereum smart contract to execute mutual authentication between fog devices using their secure keys. The main objective of this scheme was to secure fog devices from unauthorised users while maintaining user privacy. However, the initial challenges faced before the introduction of this scheme were secure communication, privacy, and validation due to the different ownership or possession of fog devices. This scheme was limited to the use of stored keys and location for device authentication purposes.

Kalaria et al. in [21] proposed a mutual authentication scheme based on fog computing that utilised one-way hash functions and Elliptic Curve Cryptography. Their security analysis highlights that their system offers protection for interacting entities against cyber-attacks such as Man-In-The-Middle and replay attacks. The proposed scheme achieved mutual authentication among fog devices; however, immutability was a problem.

The authors in [22] proposed a scheme to address the problem accompanied by the presence of trusted parties in edge computing. The problem of trusted entities registering and authenticating servers and generally the presence of this third party presented scalability challenges, and the entire network could be at risk due to threats from a single point of failure. Their solution eliminated the public trusted entity within their network framework and ended the difficulties of implementing decentralised platforms using permissioned blockchain. In this framework, authenticated users would not have to sign in to all service providers to be allowed access to the services. This scheme was limited to the use of a single authentication mechanism that could make the system vulnerable to attacks.

Maurya et al. [23] proposed an authentication technique based on a bloom filter and fuzzy extractor for IoT-enabled multi-hop Wireless Sensor Networks. This technique stopped deceitful querying of data delivery at the sensor node to stop unwanted or false flooding of data from resource-constrained IoT-enabled sensor nodes to the nearest gateway node. However, the system could put biometric data at the risk of getting stolen and reused by adversaries through biometric spoofing and exploiting the risk of a data breach in storing personal data stored in a database.

Guo and Guo [8] proposed FogHA, an anonymous handover authentication scheme that used lightweight cryptographic primitives and eliminated unnecessary authentication messages with the fog node’s assistance to achieve mutual authentication and key agreement between the fog node and mobile device. This scheme presents characteristics such as anonymity, low latency, untraceability, and resistance to compromised and insider attacks. However, an adversary could use untraceability and anonymity to its advantage to carry out attacks undetected and without a trace of the origin of the attack.

The authors in [24] proposed an identity management mechanism that maintained device anonymity and guaranteed secure negotiation of session keys and secure communication through authentication. This used blockchain technology and a secure key mechanism for authentication. Authentication was not limited to devices on a single domain; thus, devices in different administrative domains could authenticate each other without the knowledge of their identities. This characteristic could be exploited by an adversary, and this could be a possible limitation in the system.

The authors in [25] proposed a lightweight traceable device-to-device authentication and key agreement protocol based on 5G networks. This scheme generated message authorship by randomly generating a hash-based message authentication code and enabled key exchange through the Elliptic Curve. Their system delivered device traceability for network operators and anonymity to devices’ users. However, an adversary can pose as one of the devices and use brute force to generate a random hash-based message authentication code, especially in a centralised system, hereby putting the system at risk of an attack by anonymous devices.

An identity management system was proposed by [26]; this system utilised blockchain to provide non-transferability to new computed biometric attributes. These attributes are generated on each authentication attempt using a secure fuzzy extractor. This system was proposed for smart industrial organisations with the aim of procuring guaranteed identities based on anonymous biometric credentials. The use of anonymous biometric credentials could be used against this system; this could be exploited by unauthorised users and can be considered a limitation.

A decentralised system called bubbles of trust was proposed by [27]; this system was designed to identify and authenticate devices. It achieved availability and data integrity through the advantages and characteristics of blockchain technology and by using servers to create secure virtual zones where devices can verify the identities of each other. However, these virtual zones if not properly secure could serve as a point of attack for malicious devices. They identified areas that could be limitations to their approach such as its adaptability to real-time applications, the need for the initialisation phase and the evolution of cryptocurrency.

In [28], the authors presented AuthCODE, an AI-based multi-device continuous authentication architecture that maintained privacy and improved single device results by adding behavioural data generated on several devices. This offered features and characteristics that associated the communication of users with several devices and presented lists of privacy and multi-device features that harvested user communication on different devices. The use of a centralised database or trusted third party could be a limitation, and a single attack on it could collapse the system.

In [14], the authors proposed a blockchain-based biometric authentication system that processed biometric authentication through a decentralised and distributed mechanism. This also provided an auditable mechanism for the management of biometric information. This scheme with the advantages of blockchain enhanced the security of biometric data utilising through the distributed ledger, guaranteed biometric data transparency, and improved authentication.

A cancellable biometric authentication system was proposed in [29] to secure biometric data with a feature adaptive random projection. The projection metrics are generated by rendering the local feature slots and are discarded after use. This technique made the unique biometric template data unretrievable even when the user-specified key was lost. This made the user template secure and personal data protected. This scheme aimed to protect biometric data against attacks via Record Multiplicity.

The authors in [30] proposed CAB-IoT, which is a distributed and scalable continuous authentication solution based on blockchain technology. This enabled the fog nodes layer to tackle the limitations of IoT resources by processing the heavy continuous authentication-related tasks for a group of IoT devices. Their proposed system also introduced a trust module that depended on a face recognition machine learning model to control access and designed mutual authentication between end-users and fog nodes, and secure communication between the authenticated fog nodes. Their solution depended on the user’s facial data and the use of a private Ethereum network to provide privacy; however, a variety of methods can be used to manipulate users’ identities, hereby leaving this technique insecure. An authentication and authorisation framework for blockchain-enabled IoT networks was proposed in [31] using a probabilistic model. This used random numbers during authentication, which was further connected through joint conditional probability and creates a secure connection between IoT devices for data acquisition. In [32], the authors proposed a Blockchain of Things (BCoT) Gateway with the aim to enable the recording of authentication transactions in a blockchain network without changing existing device hardware or applications and introduced a new device recognition model that is appropriate for blockchain-based identity authentication, where a new feature selection method for device traffic flow was used.

Feng et al. [33] proposed an Efficient Privacy-preserving Authentication Model that utilised blockchain technology, extending its structure to preserve and achieve efficient authentication in VANETs. This supported membership verification and saved time by avoiding the verification process through a Certificate Revocation List. The authors in [9] proposed a technique that used the identity of devices as public keys during the authentication of each node by using the identity-based signature to achieve secure communications between devices on a blockchain applied system.

The schemes and systems mentioned in this section focused mainly on authentication and the use of blockchain in fog environments and IoT. Nonetheless, some schemes and systems have limitations as a result of the use of a centralised database or the use of a trusted third party. While authentication methods such as password-based authentication are considered non-practical in a decentralised system, some of the mentioned proposed schemes were not focused on user authentication in a decentralised database, and the schemes that utilised blockchain technology focused on biometric authentication and peer-to-peer communication between devices. Consequently, our proposed system is based on decentralised user authentication utilising blockchain technology and can handle various types of authentication requests with username, password, Ethereum address and user email. This addresses the secure authentication of users through a decentralised system. Table 1 summarises the existing works for decentralised, multi and mutual authentication techniques.

## 4. Proposed System

This section presents a comprehensive methodology of the proposed decentralised authentication system necessary for the authentication of users. Before presenting the proposed system in detail, we outline some of the assumptions that are considered for the development of the proposed methodology. These assumptions are commonly used in blockchain-based authentication systems such as [1]:The fog computing environment consists of mobile and immobile devices connected through several networks.The registered user devices have access to blockchain technology.The fog device must meet requirements to host the blockchain and act as a node or a server.The smart contracts should perform tasks of user registration and authentication.Nodes should not depend on other nodes to perform tasks.

We propose a decentralised authentication system using the Ethereum blockchain, smart contract, and ledger. In this system, fog nodes will serve as blockchain nodes and host a decentralised digital ledger where each node has a copy of the smart contract.

### 4.1. System Architecture

This section describes the components of the decentralised authentication system using blockchain. The architecture of the system is represented in the Figure 1. The proposed system components are described below:Ethereum Smart Contract:The contract in this authentication system is deployed to handle the task of user registration and authentication. The contract would require information such as the email, *passwrd*, and the *UserEthAdr* to enrol users upon registration and to authenticate users in the subsequent interaction with the system.Fog Node:The fog nodes are devices that act as servers and blockchain nodes; each node has a copy of the *BlockC*, *LDG*, and *SmContract*. The *BlockC* information on each node is updated when a *User* registration or authentication transaction occurs on a node. The fog device or fog server must have enough or necessary requirements to host or to be part of the *BlockCN*.Edge Devices:The end devices are user devices, and they are mapped to nodes during registration and authentication. These devices do not have the resources to host the *BlockC*.Cloud:The cloud is a large storage unit that stores, hosts and computes IoT data. This cloud server is tasked with processing and analysing data generated by IoT or edge devices.

### 4.2. Proposed System Working

In the following, we present the core functionalities of the proposed authentication system.

Initialisation:All parameters of the authentication system applied by the user during the registration are initialised by the Ethereum blockchain. These parameters are valid *UserEthAdr* (with Ethers or auth coins), valid *Usermail*, *passwrd*, and *Bdata*. The user *passwrd* and *Bdata* are hash with SHA256.User Registration:In the registration stage, the new user is required to present a valid *Usermail*, *passwrd*, and valid *UserEthAdr* to the *BlockCNet*, these data are validated and stored through the *SmContract*. The *BlockC* identifies this *User* as a valid *User*, the data provided by this user are stored on all *BlockCN*. User registration is represented in Figure 2.User Authentication:In the authentication stage, a user sends authentication requests with *Usermail*, *passwrd*, *UserEthAdr* and presents *Bdata*. The *BlockC* verifies the *User* identity through the *SmContract* and the *LDG*. The outcome of this process depends on the data provided by the *User*. The authentication is successful if *User* presents valid details; otherwise, it is declared unsuccessful. The User authentication is represented in Figure 3. In this proposed system, a new *User* registers to the network with parameters such as *Usermail*, *passwrd*, a valid *UserEthAdr*, *Username*, and *Bdata*. The data from this parameter are hashed with Secure Hash Algorithm (SHA256) and stored on the *BlockCNet*. For successful authentication, the *User* must present data according to the parameter presented by the system. In the event of inaccurate data or failed authentication, the *User* is allowed another authentication attempt.

## 5. Implementation

This section presents a description of functions in the smart contract that captures the working principle and pseudo code of the proposed authentication system.

Algorithm 1 shows the pseudocode for user registration request using the *EthAdr*, *Usermail*, *Username*, *password*, and *Bdata* of a new *User*. The *User* data are hashed and stored on the blockchain, and a new *User* is registered successfully. Figure 4 highlights the struct of the smart contract, the struct contains and defines custom data types in the *User* detail (UserDetail), these data types include the *EthAdr*, *Usermail*, *Username*, *password*, and *Bdata* of *User*. Boolean data are also used to represent scenario “isUserLoggedIn” and “true or false” binary results. The data a user must provide are included in the struct. During the registration phase, the *User* registration function is initiated to register a new *User* with data and parameters initialised by the *SmContract*; the new *User* is required to provide *Usermail*, *UserEthAdr*, *password* and *Bdata* to be registered.
**Algorithm 1** Pseudocode for user registration1:**function**user registration(EthAdr, Usermail, Username, password, BData)2:    **if** (EthAdr, Usermail, Username, password, BData = True) **then**3:        SHA256 (EthAdr, Usermail, Username, password, BData = True)4:        BlockchainUser (Store all user data and values to the Blockchain through smart contract)5:        Log (user registered)6:        **return** true7:        else8:    **end if**9:**end function**

Algorithm 2 shows the pseudocode for user authentication request using the data *EthAdr*, *Usermail*, *Username*, *password*, and *Bdata* of an existing *User*. The submitted *User* data are searched with data stored on the blockchain; if there is a match, the *User* is authenticated, or it is unsuccessful if there is no match. During authentication, the user login function requires inputs “*UserEthAdr*, *Usermail*, *password*, *Bdata*” from the *User* to validate the data presented by the *User* through SHA256 as shown in Figure 5. The boolean variable returns a true value if the *User* identity is verified or false if the data presented by the *User* are not verified. The function checkIsUserLogged (address address) checks the user’s Ethereum address to confirm the user’s status on the network and returns a boolean reply. The function logout (address address) uses the Ethereum address to end the user’s time on the network.
**Algorithm 2** Pseudocode for user authentication1:**function**user authentication(EthAdr, Usermail, Username, password, BData)2:    **if** (received authentication request from user = True) **then**3:        stored User data—match User data from blockchain through smart contract4:        **if** (User data matches data stored = true) **then**5:           Authenticate user6:           Log (“authenticated”)7:           **return** true8:           else9:           Log (“unsuccessful”) **return** false10:        **end if**11:    **end if**12:**end function**

## 6. Experimental Setup

An evaluation based on several experiments is presented to rigorously validate the proposed system. The proposed authentication system is implemented through an Ethereum smart contract utilizing Solidity. This smart contract is tested, and the simulation is run through Remix Ethereum [43]. This IDE offers various features such as the creation, deployment, testing, and debugging of smart contracts, and it permits the connection of virtual Ethereum blockchain environments such as Ganache [44] and MetaMask [45] through a web3 provider. The network layout of the system and simulations are executed in Ethereum Remix IDE and Cisco Packet Tracer. Ethereum Remix IDE is chosen for this experiment for its features which allows for the development, administering and deployment of smart contracts in a virtual blockchain environment. Cisco Packet Tracer is considered one of the best tools that deliver a network simulation environment to create virtual networks [46]. The experiment is conducted using different Ethereum accounts provided by Remix Ethereum and Ganache, each virtual Ethereum account representing a user. Multiple user registration and authentication attempts are conducted to analyse the metrics and each experiment. Data from the metrics are collected such as contract deployment cost, user registration cost, and user authentication cost. For this experiment, authentication time is not measured because security and the above-mentioned resources have been considered more important. Simulations are run through Cisco Packet Tracer to replicate the fog network with nodes and determine the time needed to send packets from user devices to the fog nodes. The simulation model is shown in Figure 6. In this, at step (1), User A presents data and sends a registration request through the edge device to the blockchain-enabled fog node. At step (2), the blockchain-enabled fog node stores user data on a distributed ledger, whereas at step (3), User A is registered as a new user. In the next step (4), User B presents data and sends an authentication request through the edge device to the blockchain-enabled fog node. At the next step (5), the blockchain-enabled fog node confirms User B’s data are valid and exist in a distributed ledger. Then, the user is authenticated or denied. The transaction cost, execution cost and miner fees are recorded for both registration and authentication requests through the blockchain network.

## 7. Performance Metrics and Results

Performance metrics and the result are presented in this section.

### 7.1. Performance Metrics

Many metrics are considered in these experiments: these are the contract deployment cost, the user registration cost, and the user authentication cost. The metrics are important for the comparison and evaluation of the proposed scheme with existing methods [1,36].

Contract deployment cost: cost of deploying the smart contract in the virtual Ethereum environment (the number of ether required to deploy the smart contract).User registration cost: the amount of ether required to register a new user in the blockchain network.User authentication cost: the amount of ether required to validate the identity of the user in the blockchain network.

### 7.2. Cisco Packet Tracer Simulation

The network is replicated in Cisco Packet Tracer [46] with fog servers (nodes) and user devices to run simulations and tests on various packets and protocols (HTTPS, SSH, SMTP, ICMP), multiple requests on wired and wireless networks to obtain the time required to send these authentication packets on the network. These packets are chosen for this experiment because they are commonly used for secure communications over computer networks [47].

## 8. Performance Evaluation

This section presents the comparison of the performance between the proposed system and the existing method [1,36] with reference to user registration and authentication. Our proposed method can be identified with the blue colour; existing methods [1,36] can be identified with orange and green colours, respectively.

The scheme [36] is designed for user authentication, utilising blockchain-enabled nodes to carry out tasks of user authentication. However, system [1] is a decentralised mutual authentication that utilised blockchain and is considered an improvement on [36]. The proposed system uses less ether (Ethereum coins) when compared to existing methods [1,36] in execution cost, transaction cost, and miner fees in both registration and authentication scenarios. The performance evaluation of the proposed system demonstrated that the system scales nicely within the increasing number of authentication requests whilst without having a significant effect on the execution time, execution cost, transaction cost and miner fees. The experiment is executed with four groups of user’s devices: 5, 10, 15, and 20 user devices. The experimental results executed in a blockchain environment for a registration and authentication scenario are discussed in this section.

### 8.1. Registration

Figure 7 displays the transaction cost, and it decreased by 30 to 40%. Figure 8 displays execution cost, and it decreased by 30 to 40%, and Figure 9 displays a miner fees decrease of 10% for a group of 5 users, 60% in the group of 10 and over 70% in the group of 15 and 20 users when compared to the metrics from the existing methods [1,36]. In our proposed system, factors such as the use of larger data values, e.g., longer password length, contributed to the increase in miner fees during user registration. The cost of registering one user through the smart contract is 698,326 ether, and the miner fee is 0.0011.

### 8.2. Authentication

Figure 10 displays the 40% decrease in transaction cost in all user groups, Figure 11 displays the 40% decrease in execution cost in all user groups and Figure 12 displays the miner fees 30% decrease in the group of 5 users, a decreased by 30% in the group of 10 users, and a decrease of over 30% and 50% in the groups of 15 and 20 users when compared to existing methods [1,36]. In our proposed system, factors such as failed authentication attempts contributed to the increase in miner fees during authentication. The cost of authenticating one user through the smart contract is 140,885 ether, and the miner fee is 0.0007.

### 8.3. Simulation Results in Cisco Packet Tracer

The results for simulation conducted in a fog computing environment in both wired and wireless networks to obtain the time required to authenticate several packets: HTTPS, SSH, SMTP, and ICMP from the user device to the fog server or blockchain node on the network is presented in Figure 13. The results showed that all the packets are delivered in 0.004 s in a wired network, while SMTP packets are delivered faster in 0.003 s in a wireless network, SSH packets are delivered in 0.009 s, and HTTPS and ICMP packets are delivered in 0.006 s. Obtained results display the time required to authenticate different packets in the proposed method; this shows our proposed method efficiently handles authentication requests quickly and can scale to multiple devices in a fog computing environment and is suited for application on edge devices.

## 9. Discussion

This section discusses possible attacks and threats on the proposed system and the advantages blockchain technology presents to mitigate these threats in an authentication system. A comparative discussion of the proposed system with existing works is also included in this section.

The principal security issues in IoT are deeply rooted in systems that provide authentication to IoT devices, services and its end users. An increase in IoT sensors and their ability to interconnect, collect and be associated with user personal data requires stringent security measures [48]. The conventional nature of the authentication system makes it less vulnerable to attacks, thereby making it robust. Utilising a combination of Ethereum address, email address, password, and data from biometric reader or device makes authentication secure. It also makes the system more resilient to common attacks targeted to authentication systems such as password attacks, Man-in-the-Middle, and spoofing. IoT presents some challenges after integration with blockchain technology. These challenges are as a result of IoT devices’ limited networking and computing capacity. The edge devices (IoT) do not have the capability to authenticate users due to their limited ability to coordinate registration and authentication; due to this, registration and authentication are performed in blockchain-enabled fog nodes/servers using edge devices as input and data collection devices.

Confidentiality is considered an important requirement in an authentication system; this is realised through the prevention of unauthorised or non-recognised access to the data in a system, network, or device. A method to achieve this is the use of encryption and hashing techniques such as SSL and SHA256. The proposed system utilised SHA256 to hash user’s data; this transforms the data into a secure unreadable format that is readable without a secret key. Similar to confidentiality, integrity is an important characteristic of a good access-controlled or access restricted environment. This is necessary to mitigate or evade the modification or tampering of data to make the system secure against attacks such as replay and Man-in-the-Middle attacks. The encryption of messages mitigates these attacks.

The use of blockchain technology in authentication system without relying on a third party makes the proposed system robust and resistant to attacks. Blockchain generally provides high-level security for authentication systems and the fog computing environment [49]. This high-level security encompasses data confidentiality through secure encryption, data integration through credibility and traceability, data availability through distributed and remote access, access control through user authorisation, trust through secure communication and mitigation of attacks and preserves user identity through data and identity privacy [50]. Our proposed system depends on the unique properties of blockchain technology for security. This mitigates vulnerabilities in SSH. Although SSH has its advantages and provides robust authentication, however, it does not provide traceability or trust and is vulnerable to attacks such as connection hijacking, Man-in-the-Middle attack, IP spoofing and connection hijacking [51]. The use of blockchain technology would mitigate threats and provide security, traceability, trust and anonymity, and the proposed system’s reliance on blockchain technology makes it robust against common authentication attacks.

The use of blockchain technology makes the proposed system robust against Denial-of-Service attacks (DoS) by storing user data, mappings, and other records in a distributed and decentralised Ethereum ledger. Blockchain’s provision of anonymity and privacy makes authentication secure and offers security for sensitive user data. The resilience and global distribution of Ethereum’s public ledger make it resistant to several attacks (DoS inclusive)—the ledger is shared and protected by various nodes where these data and records are hosted with integrity. Generally, the unique characteristics of blockchain technology provide extra security measures to authentication systems, IoT, and cloud devices.

Possible vulnerabilities in smart contract code can pose a risk to the system and users of the contract. Vulnerabilities in Solidity open up a possibility to take over control of untrusted functions in another smart contract; this is commonly known as a re-entrancy attack. During this attack, smart contract 1 calls an unspecified function from smart contract 2, which could give way to call a function from smart contract 2 with a malicious aim. The proposed system addresses this concern by deploying smart contracts with absolute necessary functionalities (code) and removing auxiliary functions.

The evaluation results have demonstrated that the proposed system is fully scalable to multiple IoT devices and can still carry out the tasks of user registration and authentication. Results also highlighted that the proposed system used less resources in Ethereum (ether) when compared to existing methods [1,36].

## 10. Conclusions

In this paper, a decentralised user authentication system was designed and implemented using Ethereum smart contract. It aimed to address user authentication in a decentralised environment and address fog computing security problems inherited from IoT and cloud computing. It provided a solution for immutability, scale-ability problems in fog computing and proposed the use of conventional authentication in a fog and a blockchain environment to achieve the distributed authentication system. Compared to other methods, the proposed system utilised commonly used authentication factors to ensure frictionless adoption in real-world applications. Experimental results revealed that the proposed method achieved performance improvement, used less resources in Ethereum (ether) and could be scaled to more devices and relatively used less resources when compared to existing methods. In addition to this, the comparison of results with existing methods showed that the proposed authentication system consumed less gas. During user registration, the transaction cost decreased by 30% to 40%, execution cost decreased by 40%, and miner fees decreased by 10% for the group of 5 users, 60% in the group of 10, and over 70% in the group of 15 and 20 users when compared to the metrics from the existing methods. During user authentication, there was a 40% decrease in transaction cost in all user groups, 40% decrease in execution cost in all user groups, and 30% decrease in miner fees in the group of 5 users, 30% decrease in the group of 10 users, over 30% and 50% decrease in the groups 15 and 20 users when compared to existing methods. Future direction would focus on replicating the experiment using another approach such as NEO smart contracts to improve authentication.

## Figures and Tables

**Figure 1 sensors-22-03956-f001:**
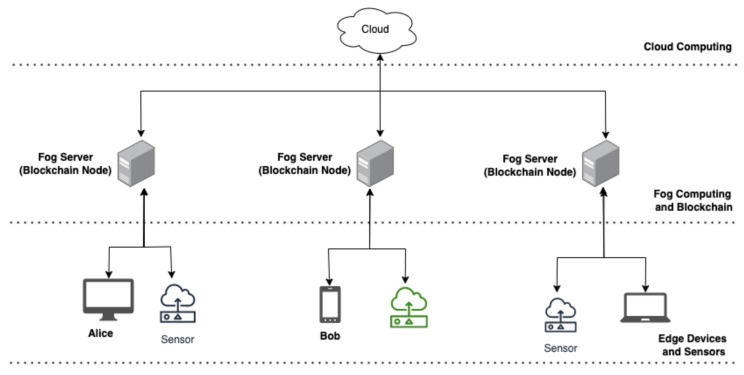
System architecture for fog-enabled blockchain-based authentication system.

**Figure 2 sensors-22-03956-f002:**
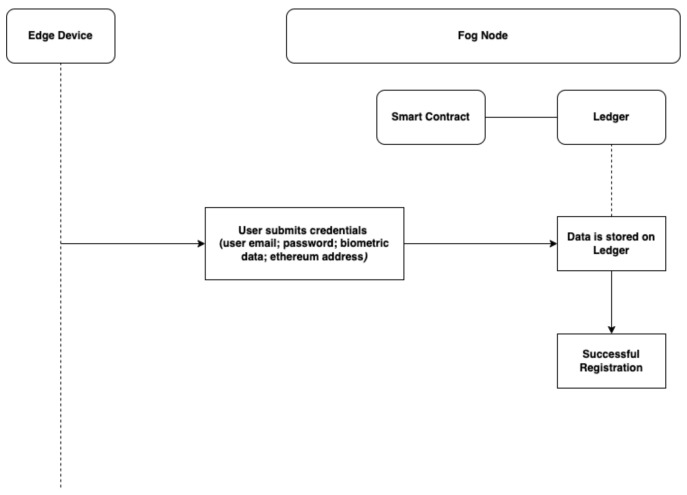
Flow diagram for user registration.

**Figure 3 sensors-22-03956-f003:**
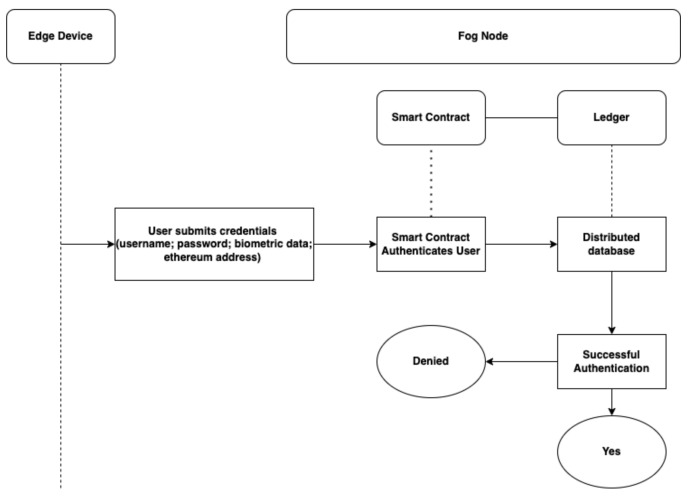
Flow diagram for user authentication.

**Figure 4 sensors-22-03956-f004:**
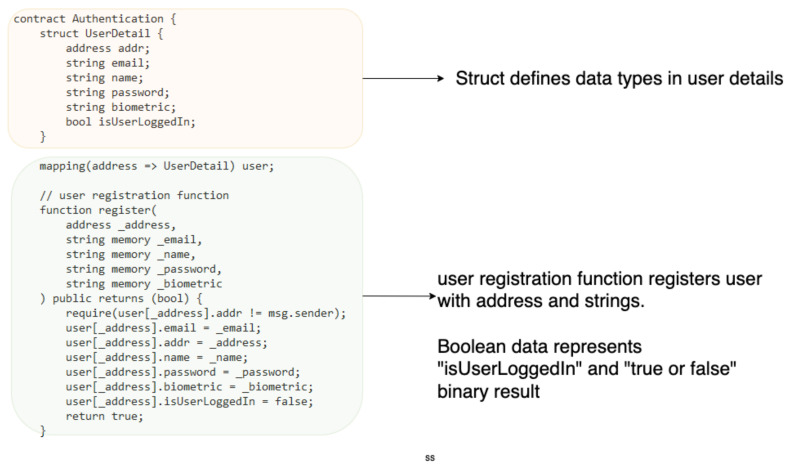
An example of smart contract code for user registration.

**Figure 5 sensors-22-03956-f005:**
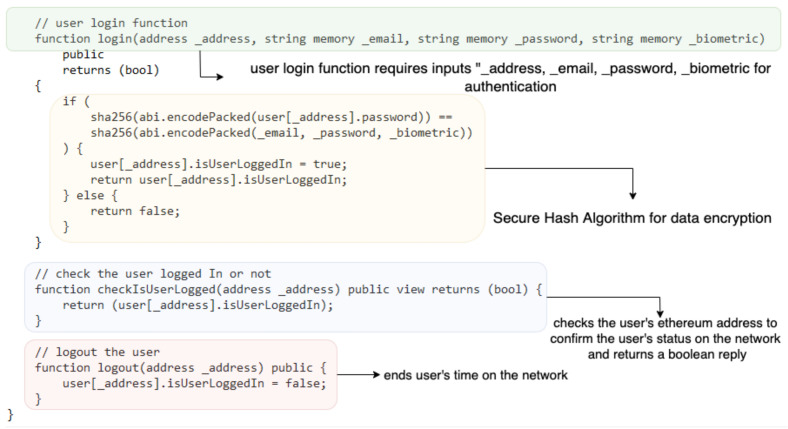
An example of smart contract code for user validation.

**Figure 6 sensors-22-03956-f006:**
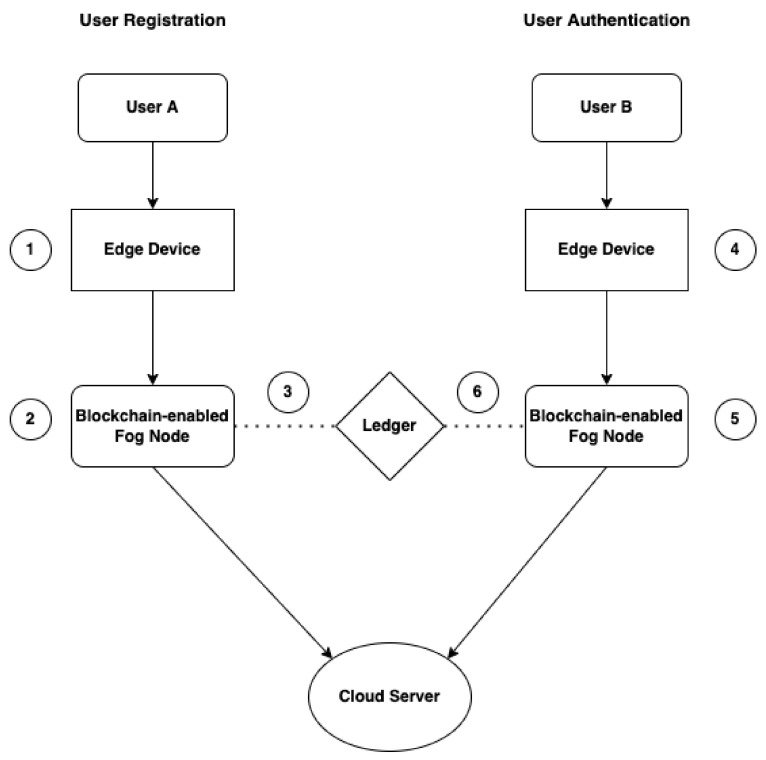
The simulation model.

**Figure 7 sensors-22-03956-f007:**
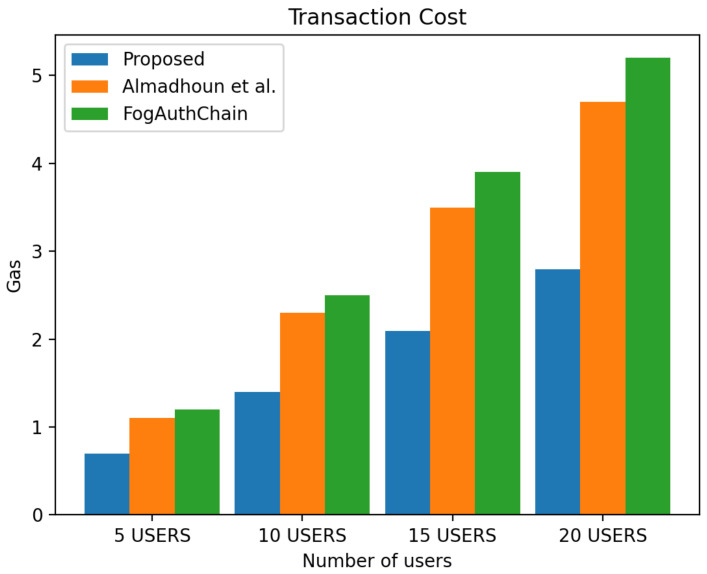
Comparison of experimental results of the proposed system and existing systems (Almadhoun et al. [36] and FogAuthChain [1]) on registration transaction cost.

**Figure 8 sensors-22-03956-f008:**
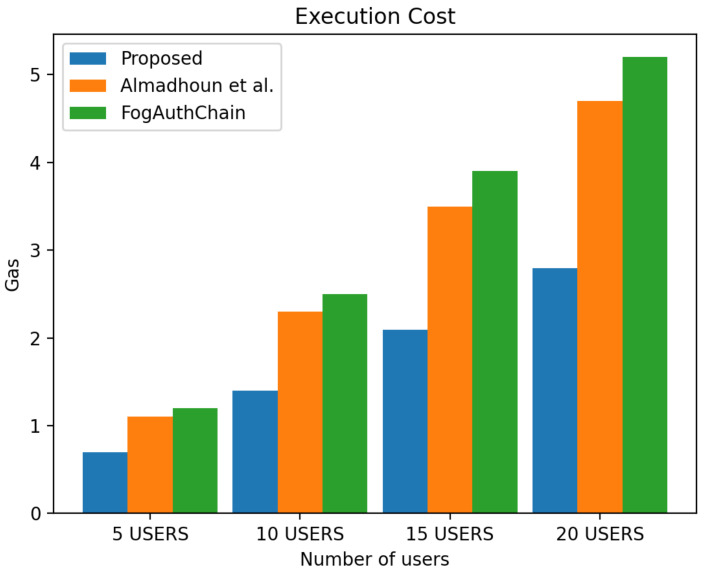
Comparison of experimental results of the proposed system and existing systems (Almadhoun et al. [36] and FogAuthChain [1]) on registration execution cost.

**Figure 9 sensors-22-03956-f009:**
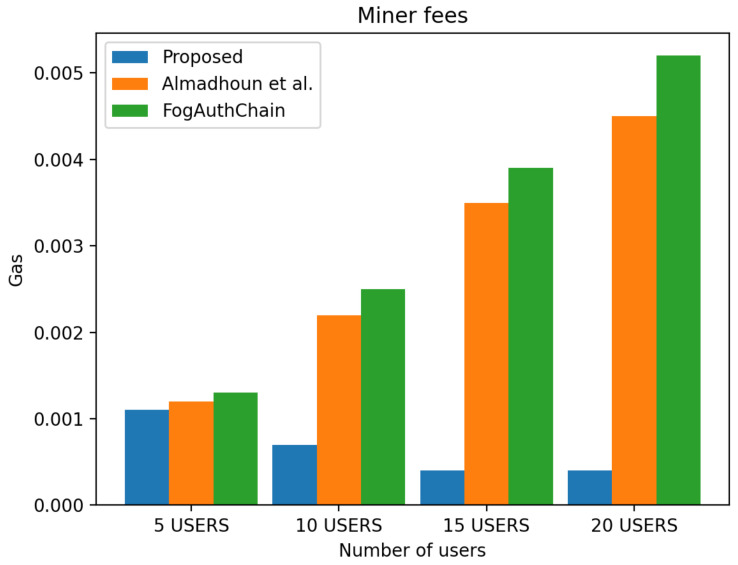
Comparison of experimental results of the proposed system and existing systems (Almadhoun et al. [36] and FogAuthChain [1]) on registration miner fees.

**Figure 10 sensors-22-03956-f010:**
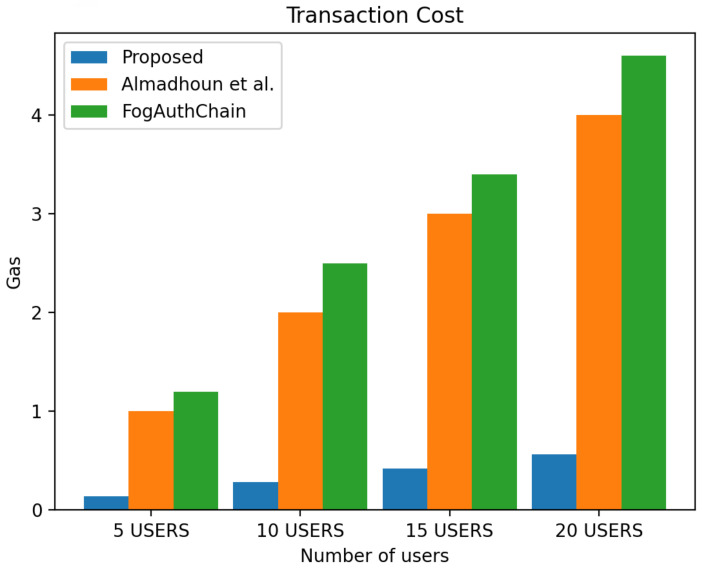
Comparison of experimental results of the proposed system and existing systems (Almadhoun et al. [36] and FogAuthChain [1]) on authentication transaction cost.

**Figure 11 sensors-22-03956-f011:**
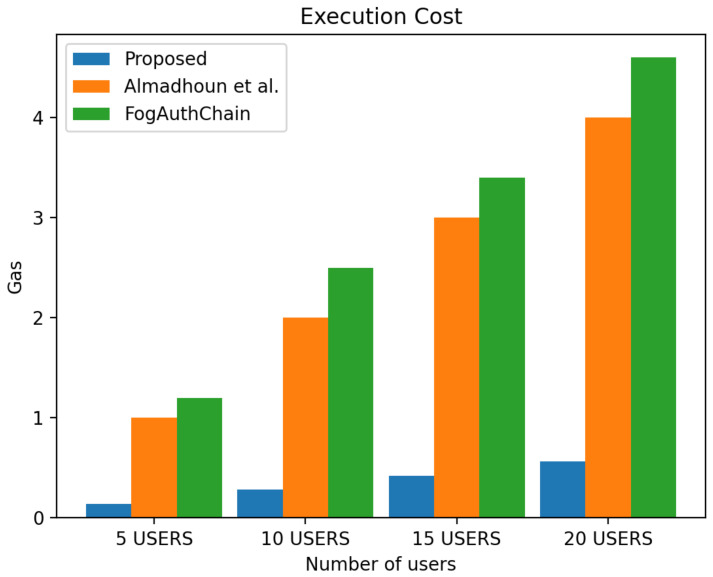
Comparison of experimental results of the proposed system and existing systems (Almadhoun et al. [36] and FogAuthChain [1]) on authentication execution cost.

**Figure 12 sensors-22-03956-f012:**
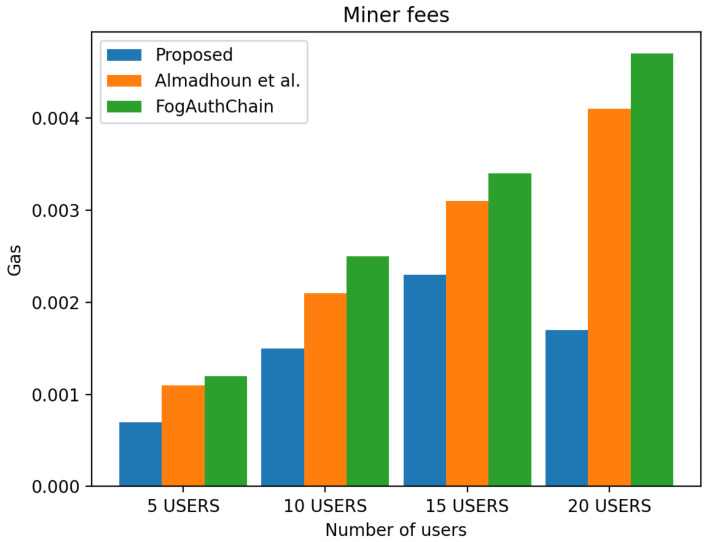
Comparison of experimental results of the proposed system and existing systems (Almadhoun et al. [36] and FogAuthChain [1]) on authentication miner fees.

**Figure 13 sensors-22-03956-f013:**
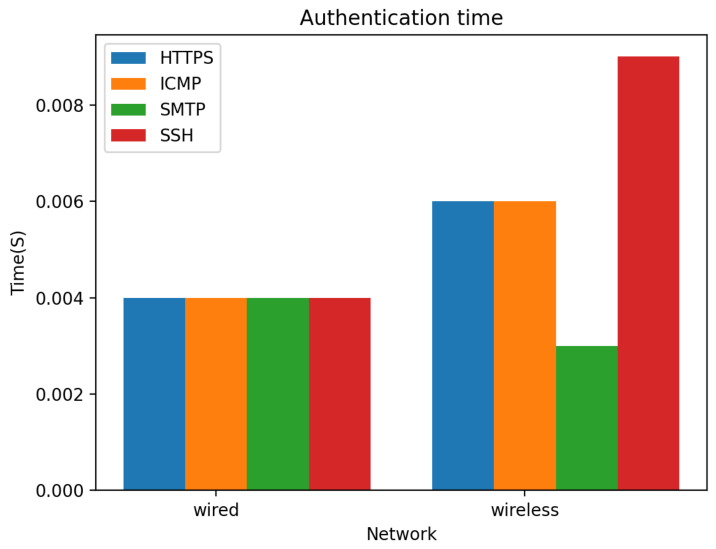
Simulation results showing the time required for authentication and sending packets.

**Table 1 sensors-22-03956-t001:** Comparative analysis of existing work for decentralised, multi, and mutual authentication whilst considering the scalability and robustness of the authentication systems and frameworks (Y = Yes, N = No, NA = Not Mentioned). (DM = Distributed model, DBP = Data breach protection, Mutual Auth = Mutual authentication, MFA = Multi-factor authentication, S = Scalability, RAA = Robust against attacks).

Work	Distributed Model	Data Breach Protection	Mutual Authentication	Multi-Factor Authentication	Scalability	Robust against Attacks
FogAuthChain [1]	Y	Y	Y	N	Y	Y
FogHA [8]	Y	Y	Y	N	NA	Y
Blockchain meets IoT [11]	Y	N	N	N	Y	NA
B. Gupta [34]	Y	NA	Y	Y	NA	NA
SDFC [35]	Y	NA	N	N	NA	NA
Esposito et al. [19]	Y	NA	N	N	NA	NA
Khalid et al. [20]	Y	NA	Y	N	NA	Y
Kalaria et al. [21]	Y	Y	Y	N	NA	Y
Dechain [22]	Y	Y	Y	N	NA	Y
Meng et al. [24]	Y	NA	Y	N	NA	NA
Chow and Ma [25]	N	NA	Y	N	NA	Y
Bubble of trust [27]	Y	Y	Y	N	Y	Y
AuthCODE [28]	N	NA	N	N	NA	NA
Almadhoun et al. [36]	Y	Y	Y	N	Y	Y
Leandrloffi et al. [37]	N	Y	Y	Y	NA	Y
Masfog [38]	Y	Y	Y	NA	NA	NA
FogBus [7]	Y	NA	N	N	Y	NA
DA-SADA [39]	Y	Y	NA	N	NA	Y
PF-BTS [40]	Y	Y	NA	N	Y	Y
AttriChain [41]	Y	NA	NA	N	NA	Y
Patil et al. [42]	Y	NA	NA	N	NA	Y
Our Proposed work	Y	Y	Y	Y	Y	Y

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
