# Peer review of "Securing Fog Computing with a Decentralised User Authentication Approach Based on Blockchain"

_sensors, 2022, doi:10.3390/s22103956_

Round 1

Reviewer 1 Report

My opinion is that the paper is not well written and organized. There are many general remarks because the whole paper is not well formatted and edited:

  1. There is a lot of ambiguity in the understanding of the text even in the abstract, for example sentences in lines 4, 5 and 6.
  2. The text is not well edited, for example lines 340 and 341. There are too many smaller subchapters in Chapter 4 that are not very clear.
  3. The figures in Chapter 4 are too large.
  4. The titles of the figures in Chapter 8 do not match the content that the figures display.

In addition, there are serious shortcomings related to paper quality:

  1. Chapter 5. Implementation does not provide enough information about the implementation itself. The data structure is shown in Figure 4, but without the necessary definition of individual data. The code in Figure 5 is unclear. Proposal, define the data structure and display the code as pseudocode.
  2. Chapters 6. Experimental setup and 7. Performance metrics and Results don't provide enough information. It is necessary to present the used simulation model and explain it.
  3. In Chapter 8. Performance Evaluation all the results obtained are not well explained. Certain explanations are missing, for example why the results in Figure 11 for 10 users differ from other results (large deviation of the proposed system).
  4. The conclusion is not clearly written and it is difficult to understand the advantages of the proposed system.

Author Response

  • Reviewer#1 Minor issues - reviewer # 1, Concern # 1: There is a lot of ambiguity in the understanding of the text even in the abstract, for example sentences in lines 4, 5 and 6.

Author response: Thank you for highlighting this, we have now updated the text.

Author action: Lines 2, 3,  4, 5 and 6 are rewritten in the abstract. 

  • Reviewer#1 Minor issues - reviewer # 1, Concern # 2: The text is not well edited, for example lines 340 and 341. There are too many smaller subchapters in Chapter 4 that are not very clear.

Author response: Thank you for your insightful comment, we have now updated the text, and removed third level subsections.

Author action: Lines 340 and 341 are rewritten, and can now be found in lines 362 and 364. Relatively smaller third-level subsections 4.1.1, 4.1.2, 4.1.3, 4.1.4, 4.2.1, 4.2.2, and 4.2.3 are rewritten to first-level bullet points.

  • Reviewer#1 Minor issues - reviewer # 1, Concern # 3: The figures in Chapter 4 are too large.

Author response: Thank you for highlighting this, we have now updated image sizes in Chapter 4.

Author action: The width of each Image in Chapter 4 is reduced from 15 cm to 10 cm.

  • Reviewer#1 Minor issues - reviewer # 1, Concern # 4: The titles of the figures in Chapter 8 do not match the content that the figures display.

Author response: Thank you for highlighting this, we have now updated and reviewed the titles of Figures 7, 8, 9, 10, 11, 12, and 13.

Author action: We have corrected the mistakes in the title of Figures in Section 8.

  • Reviewer#1 Major issues - reviewer # 1, Concern # 1: Chapter 5. Implementation does not provide enough information about the implementation itself. The data structure is shown in Figure 4, but without the necessary definition of individual data. The code in Figure 5 is unclear. Proposal, define the data structure and display the code as pseudocode.

Author response: We appreciate the feedback from the review, we have added pseudocode as proposed.

Author action: Pseudocode is added to the implementation section and more explanation is added in Section 5.

  • Reviewer#1 Major issues - reviewer # 1, Concern # 2: Chapters 6. Experimental setup and 7. Performance metrics and Results don't provide enough information. It is necessary to present the used simulation model and explain it.

Author response: We appreciate the constructive feedback to provide more information and present the used simulation model.

Author action: Added a Figure 6 of the simulation model and text to explain the model, and gave more explanation of the results.

  • Reviewer#1 Major issues - reviewer # 1, Concern # 3: In Chapter 8. Performance Evaluation all the results obtained are not well explained. Certain explanations are missing, for example why the results in Figure 11 for 10 users differ from other results (large deviation of the proposed system).

Author response: Thank you for your valuable feedback, the large deviation from miner fees in the proposed system in Figure 11 is a result of a mistake while plotting the chart.

Author action: Plotting error in Figure 11 has been fixed now. Other paragraphs are expanded to give more explanation for the performance evaluation.

  • Reviewer#1 Major issues - reviewer # 1, Concern # 4: The conclusion is not clearly written and it is difficult to understand the advantages of the proposed system

Author response: Thank you for your valuable feedback.

Author action: Parts of the conclusion are rewritten to further explain the advantages of the proposed system.

Reviewer 2 Report

In this paper, the authors present a blockchain-based authentication approach for fog computing. The proposed system is focused on user authentication that utilizes a combination of a username, password, biometric data, email address, and Ethereum address.

While the proposed approach holds merit, there are several aspects that need to be addressed before the paper is ready for publication.

Major Comments:
1. Related works missed a few significant works addressing the same problem, such as:

Al-Naji, F.H., Zagrouba, R.: Cab-iot: continuous authentication architecture based on blockchain for internet of things. Journal of King Saud University-Computer and Information Sciences (2020)

Hammi, M.T., Hammi, B., Bellot, P., Serhrouchni, A.: Bubbles of trust: A decentralized blockchain-based authentication system for iot. Computers & Security 78, 126–142 (2018)

Gong, L., Alghazzawi, D.M., Cheng, L.: Bcot sentry: A blockchain-based identity authentication framework for iot devices. Information 12(5), 203 (2021)

Tahir, M., Sardaraz, M., Muhammad, S., Saud Khan, M.: A lightweight authentication and authorization framework for blockchain-enabled iot network in health-informatics. Sustainability 12(17), 6960 (2020)

2. The discussion section does not compare the obtained results with previous works. I suggest adding a table that compares the results in terms of attack-resistance, computational cost, computation time, and communication overhead. the feature comparison in Table 1 is useful. However, it doesn't cover the simulation results.

3. The paper title is misleading. It does not mention user authentication. It sounds like fog nodes authenticating each other. I suggest altering the title to a name indicative of user-authentication within an IoT/fog environment.

4. Add a small part at the end of the introduction that explains the layout of the rest of the paper.

Minor Comments:
1. The paper needs language review to ensure that it is easy to read.

Author Response

  • Reviewer#2 Minor issues - reviewer # 2, Concern # 1: The paper needs language review to ensure that it is easy to read.

Author response: Thank you for your insightful comment on the manuscript.

Author action: The entire paper has been reviewed with a fresh pair of eyes by the authors - spelling mistakes and other errors have been corrected.

  • Reviewer#2 Major issues - reviewer # 2, Concern # 1: Related works missed a few significant works addressing the same problem, such as:
    • Al-Naji, F.H., Zagrouba, R.: Cab-iot: continuous authentication architecture based on blockchain for internet of things. Journal of King Saud University-Computer and Information Sciences (2020)
    • Hammi, M.T., Hammi, B., Bellot, P., Serhrouchni, A.: Bubbles of trust: A decentralized blockchain-based authentication system for iot. Computers & Security 78, 126–142 (2018)
    • Gong, L., Alghazzawi, D.M., Cheng, L.: Bcot sentry: A blockchain-based identity authentication framework for iot devices. Information 12(5), 203 (2021)
    • Tahir, M., Sardaraz, M., Muhammad, S., Saud Khan, M.: A lightweight authentication and authorization framework for blockchain-enabled iot network in health-informatics. Sustainability 12(17), 6960 (2020)

Author response: Thank you for the reviewer to highlight and recommending missing significant works.

Author action: Recommended significant works are added in the manuscript between lines 306 and 323. Significant work “bubbles of trust” was already included in related work from line 279 to 286.

  • Reviewer#2 Major issues - reviewer # 2, Concern # 2: The discussion section does not compare the obtained results with previous works. I suggest adding a table that compares the results in terms of attack-resistance, computational cost, computation time, and communication overhead. the feature comparison in Table 1 is useful. However, it doesn't cover the simulation results.

Author response: Thank you for your valuable feedback.

Author action: A brief comparison was added to the discussion section.

  • Reviewer#2 Major issues - reviewer # 2, Concern # 3: The paper title is misleading. It does not mention user authentication. It sounds like fog nodes authenticating each other. I suggest altering the title to a name indicative of user-authentication within an IoT/fog environment.

Author response: Thank you for your valuable feedback. We agree the title should reflect user authentication.

Author action: Based on the reviewer’s constructive comment, the title of the paper has been updated to “Securing Fog Computing with a Decentralised User Authentication Approach based on Blockchain”.

  • Reviewer#2 Major issues - reviewer # 2, Concern # 4: Add a small part at the end of the introduction that explains the layout of the rest of the paper.

Author response: We appreciate your valuable insight.

Author action: Manuscript layout is added at the end of Section 1.

Reviewer 3 Report

The manuscript proposed an authentication system for IoT fog networks based on blockchain and smart contracts. This reviewer has two main concerns:

- the proposed authentication protocol has been claimed to be secure against various attacks compared to previous work, as discussed in table 1. however, no concrete security analysis has been provided. please include analysis / proof of security against the attacks.

- for the packet tracer simulation in section 8.3 and figure 12, various secure communication protocols such as HTTPS, SSH, etc are considered. if the network is already using SSH which provides robust authentication then what is the need for another blockchain-based authentication approach along with that?

Author Response

  • Reviewer#3 Major issues - reviewer # 3, Concern # 1: the proposed authentication protocol has been claimed to be secure against various attacks compared to previous work, as discussed in table 1. however, no concrete security analysis has been provided. please include analysis / proof of security against the attacks.

Author response: Thank you for your valuable feedback.

Author action: Provided more details about the security of the proposed system based on the unique properties of blockchain in section 9.

  • Reviewer#3 Major issues - reviewer # 3, Concern # 2: for the packet tracer simulation in section 8.3 and figure 12, various secure communication protocols such as HTTPS, SSH, etc are considered. if the network is already using SSH which provides robust authentication then what is the need for another blockchain-based authentication approach along with that?

Author response: Thank you for your valuable insight and highlighting a very relevant concern.

HTTPS and SSH have their advantages and provide robust authentication. Both of these techniques rely on a public-key cryptosystem, which is a centralized approach for authentication (primarily because of involving a trusted third party).  We have included this use-case to keep research open for the future where blockchain has the capability to replace public certificates signed by a trusted third party. Blockchain technology can be used to create its own public/private key pair and use these in HTTPS and SSH protocols in a decentralized manner. Also, we believe that blockchain technology can be used to further enhance the robustness of HTTPS and SSH authentication by adding a decentralization approach for validating the digital certificates.

Author action: Provided explanation in the response letter.

Round 2

Reviewer 1 Report

I think that the authors have invested extra effort and taken into account all the comments of the reviewer, and the paper is now at a satisfactory level.

Reviewer 2 Report

I would like to thank the authors for addressing my concerns. I have no further comments.

Reviewer 3 Report

The authors have revised the manuscript to address previous concerns.